# Metabolic Risk Factors for Hepatocellular Carcinoma in Patients with Nonalcoholic Fatty Liver Disease: A Prospective Study

**DOI:** 10.3390/cancers14246234

**Published:** 2022-12-17

**Authors:** Samuel O. Antwi, Emily C. Craver, Yvonne A. Nartey, Kurt Sartorius, Tushar Patel

**Affiliations:** 1Division of Epidemiology, Department of Quantitative Health Sciences, Mayo Clinic, Jacksonville, FL 32224, USA; 2Division of Clinical Trials and Biostatistics, Department of Quantitative Health Sciences, Mayo Clinic, Jacksonville, FL 32224, USA; 3Department of Internal Medicine and Therapeutics, School of Medical Sciences, University of Cape Coast, Cape Coast 03321, Ghana; 4School of Laboratory Medicine and Molecular Sciences, College of Health Sciences, University of Kwazulu-Natal, Durban 04013, South Africa; 5UKZN Gastrointestinal Cancer Research Unit, University of Kwazulu-Natal, Durban 04013, South Africa; 6Department of Transplantation, Mayo Clinic, Jacksonville, FL 32224, USA

**Keywords:** liver cancer, HCC, hepatocellular carcinoma, risk factors, diabetes, obesity, metabolic syndrome

## Abstract

**Simple Summary:**

About 30% of Americans have nonalcoholic fatty liver disease (NAFLD), and some of these individuals may develop hepatocellular carcinoma (HCC), a frequently fatal cancer. A major current challenge is how to identify those NAFLD patients who are likely to progress to HCC. Metabolic conditions such as obesity and diabetes may drive the progression of NAFLD to HCC, but the extent of risks associated with these conditions is unknown. The aim of this study was to assess the magnitudes of risk and population-attributable risk fractions associated with various metabolic conditions regarding HCC risk. Our study shows that NAFLD patients with diabetes have the highest risk for developing HCC, followed by those with metabolic syndrome and then obesity. Because the incidence of NAFLD is expected to increase further, these results are important for designing targeted strategies to prevent HCC development in NAFLD patients with high-risk metabolic conditions.

**Abstract:**

Non-alcoholic fatty liver disease (NAFLD) is a fast-growing public health problem and predisposes to hepatocellular carcinoma (HCC) in a significant proportion of patients. Metabolic alterations might underlie the progression of NAFLD to HCC, but the magnitudes of risk and population-attributable risk fractions (PAFs) for various metabolic conditions that are associated with HCC risk in patients with NAFLD are unknown. We investigated the associations between metabolic conditions and HCC development in individuals with a prior history of NAFLD. The study included 11,245 participants in the SEER-Medicare database, comprising 1310 NAFLD-related HCC cases and 9835 NAFLD controls. We excluded individuals with competing liver diseases (e.g., alcoholic liver disease and chronic viral hepatitis). Baseline pre-existing diabetes mellitus, dyslipidemia, obesity, hypertension, hypothyroidism, and metabolic syndrome were assessed. Multivariable-adjusted logistic regression was used to calculate odds ratios (ORs) and 95% confidence intervals (CIs). PAFs were also calculated for each metabolic condition. The results show that diabetes (OR = 2.39, 95% CI: 2.04–2.79), metabolic syndrome (OR = 1.73, 95% CI: 1.49–2.01), and obesity (OR = 1.62, 95% CI: 1.43–1.85) were associated with a higher HCC risk in individuals with NAFLD. The highest PAF for HCC was observed for pre-existing diabetes (42.1%, 95% CI: 35.7–48.5), followed by metabolic syndrome (28.8%, 95% CI: 21.7–35.9) and obesity (13.2%, 95% CI: 9.6–16.8). The major predisposing factors for HCC in individuals with NAFLD are diabetes mellitus, metabolic syndrome, and obesity, and their control would be critically important in mitigating the rising incidence of NAFLD-related HCC.

## 1. Introduction

Nonalcoholic fatty liver disease (NAFLD) is a growing global public health problem [1,2,3]. NAFLD is a metabolic disorder that is often defined by the accumulation of triglycerides in the liver in the absence of viral hepatitis, alcoholic liver disease, autoimmune hepatitis, and other rare genetic syndromes that predispose to chronic liver disease [4,5,6,7]. NAFLD may progress to liver cirrhosis, primary liver cancer, or end-stage liver disease requiring liver transplantation [5,7]. The continuum of NAFLD progression to cirrhosis and HCC often occurs in the background of necroinflammation, fibrosis, regenerative nodules, and changes in the liver vasculature [2,8,9,10,11]. Currently, NAFLD is the second leading indication for liver transplantation among American men and women combined (after alcohol abuse) and is the leading cause of liver transplantation among American women [12,13]. NAFLD is also the fastest-growing cause of primary liver cancer (hepatocellular carcinoma, HCC) among Americans [2,14]. Importantly, NAFLD-related HCC cases are expected to increase further in the United States (US) owing to the increasing prevalence of obesity and reduction in the prevalence of chronic viral hepatitis C infection (HCV) due to the widespread use of the highly effective direct-acting antiviral medications for the treatment of HCV and vaccinations for hepatitis B virus (HBV) infection [8,14,15]. However, not all patients with NAFLD will progress to the more deleterious liver pathologies, such as HCC, which is a frequently fatal cancer [2,14]. There is, therefore, an urgent need to identify those patients with NAFLD who are likely to progress to the more lethal HCC for active surveillance to enhance early detection. 

The risk factors for HCC development in persons with NAFLD are not entirely clear, but the progression of NAFLD to HCC has been associated with the compounding presence of additional metabolic conditions, such as diabetes mellitus and dyslipidemia [2,5,16]. As an example, diabetes is a risk factor for HCC [17,18,19], and studies have shown that diabetes often co-exists with NAFLD, with up to 75% of NAFLD patients having a concurrent diagnosis of diabetes [16,20], and the temporal relationship between diabetes and NAFLD is less clear [21,22]. However, the presence of diabetes in NAFLD patients has been associated with a higher HCC risk [23,24]. Furthermore, the relative contributions of various metabolic conditions to the development of HCC in patients with NAFLD, in the presence or absence of liver cirrhosis, is not completely known. There is also a critical need to determine the proportion of HCC cases that are attributable to each of these metabolic risk factors to inform public health strategies for controlling or halting the increasing incidence of NAFLD-related HCC [3,14]. Determining the primary causes of HCC in patients with NAFLD will also inform clinical decisions about HCC screening in clearly defined high-risk groups. 

The primary aim of this study was to assess the magnitudes of risk and population-attributable risk fractions (PAFs) associated with various metabolic conditions in relation to HCC risk. The secondary aim was to assess risk magnitudes and PAFs for various metabolic conditions by sex and liver cirrhosis status since the pathological progression of NAFLD to HCC may occur in the presence or absence of liver cirrhosis [7,8,25]. 

## 2. Materials and Methods

### 2.1. Study Setting and Data Source

Data were obtained from the U.S. National Cancer Institute’s Surveillance Epidemiology and End Results-Medicare matched program (SEER-Medicare). The design and methods used in the SEER-Medicare program have been described [26,27,28,29]. In brief, the SEER-Medicare data are curated through the linkage of data from the US National Cancer Institute’s SEER program and Medicare enrollment and claims files. The SEER data component used for this study comprised twenty-one registries (SEER-21, 2003–2017) consisting of twelve states with central cancer registries (Alaska, Connecticut, Hawaii, Idaho, Iowa, Kentucky, Louisiana, Massachusetts, New Jersey, New Mexico, New York, and Utah) and eight registries from metropolitan areas (Greater California, Los Angeles, San Francisco-Oakland, San Jose-Monterey, Seattle-Puget Sound, Greater Georgia, Rural Georgia, Metropolitan Atlanta, and Metropolitan Detroit). However, we did not have any data from Alaska or Rural Georgia after the exclusions described below. Collectively, the SEER-21 registries represent about 36.7% of the United States population [30]. 

The Medicare program is a government-administered national health insurance program that provides coverage for about 97% of Americans aged 65 years or older and individuals with disability or end-stage renal disease [31]. Virtually all Medicare beneficiaries (~99%) enroll in Medicare Part A, which provides coverage for healthcare services, such as in-patient hospital care, skilled-nursing facility care, some home healthcare, and hospice care. Ninety-six percent of those participating in Medicare Part A also participate in Medicare Part B through payment of monthly premiums to cover the cost associated with outpatient care, medical supplies, and some preventive services [31]. The Medicare Parts A and B files contain dates when various Medicare services were provided to enrollees as well as the International Classification of Diseases, ninth and tenth revisions, Clinical Modification (ICD-9-CM and ICD-10-CM) codes for medical diagnosis and procedures, and codes for all billed claims. 

To construct the SEER-Medicare data file, individual-level data with identifiers on all persons diagnosed with cancer in the SEER database are linked to individual-level Medicare master enrollment files to obtain medical information on all persons enrolled in the Medicare program [27]. About 93% of Medicare-eligible individuals aged 65 years or older in the SEER files have been matched to the Medicare enrollment files [27]. 

### 2.2. Study Design

We performed a nested case–control study using prospective data from the SEER-Medicare program. HCC cases were identified from the SEER cancer registry files, and population-based cancer-free controls were identified from the same geographic regions of the SEER cancer registries [29]. Following approval by the Mayo Clinic Institutional Review Board (IRB), we investigated the magnitudes of HCC risk and PAFs associated with each of the following cardiometabolic conditions: obesity, dyslipidemia, hypertension, hypothyroidism, type II diabetes mellitus, and metabolic syndrome. PAF reflects the number of HCC cases that could be theoretically eliminated should a particular metabolic risk factor be eliminated from the population. 

### 2.3. Study Population and Sample Selection

As part of the inclusion criteria, case participants were required to be 65 years or older and have a primary-site, histologically confirmed diagnosis of HCC based on the World Health Organization’s classification [32]. We used the International Classification of Diseases for Oncology, version 3 (ICD-O-3) topography code C22.0 and morphology codes 8170–1875 to select the HCC cases [25,33,34,35,36]. We then restricted the cases to persons with the primary diagnosis of HCC and who were continuously enrolled in Medicare Parts A and B at least 3 years before the diagnosis of HCC to allow for sufficient time to assess baseline characteristics, including a prior diagnosis of NAFLD. This resulted in a minimum age of 68 years for all participants. Because Medicare eligibility starts at 65 years of age, we excluded persons younger than 65 years at the time of HCC diagnosis, persons enrolled in Medicare because of end-stage renal disease or disability, persons with HCC diagnosed solely by autopsy report or death certificate alone, and persons jointly enrolled in Medicare and other health-maintenance organizations (HMOs). The HMO exclusions were performed to avoid the misclassification of patients whose claims information may have been reported to the HMO but not to Medicare. Because the study is focused on NAFLD-related HCC, we also excluded participants who did not have a diagnosis of NAFLD at least 3 years before HCC diagnosis. NAFLD was identified using the ICD-9-CM code 571.8 and ICD-10-CM code K76.0 [33]. To optimally control for confounding by other liver diseases and known risk factors of HCC, we further excluded individuals with a history of viral hepatitis B or C, alcoholic liver disease, alcohol use disorder, autoimmune hepatitis, primary biliary cirrhosis (PBC), primary sclerosing cholangitis (PSC), hemochromatosis, Wilson’s disease, alpha-1-antitrypsin deficiency, Budd-Chiari syndrome, unspecified chronic hepatitis, and secondary or unspecified biliary cirrhosis, using expert-consensus recommended ICD codes [33], as shown in Appendix A. 

Cancer-free controls were selected from a 5% random sample of Medicare beneficiaries residing in the same geographic regions of the SEER cancer registries included in this study. The controls were subjected to the same inclusion and exclusion criteria as the cases. Specifically, the controls had to be continuously enrolled in Medicare Parts A and B for at least 3 years, could not be younger than age 68 years on 31 December 2017 or have died before age 68 years, must have a diagnosis of NAFLD, must not be jointly enrolled in Medicare and HMO, and must not have been enrolled in Medicare because of a diagnosis of end-stage renal disease or disability. We assigned the controls a pseudo-diagnosis date using a random number generator and matched the controls to cases on the year of search for baseline information and metabolic risk factors to minimize the impact of changes in diagnosis trends over time. Additionally, the controls were required to live in the SEER registry area by the pseudo-diagnosis date. As with the cases, we excluded potential controls with the liver diseases listed above. 

During the study period (2003–2017), we identified 42,954 HCC cases in individuals 68 years of age or older at diagnosis with at least 3 years of prior medical information in the Medicare claims files, without enrollment in HMO, and not enrolled in Medicare because of end-stage renal disease or disability. We excluded those with non-primary HCC (*n* = 37), those with mixed HCC and intrahepatic cholangiocarcinoma (*n* = 23), those identified by death certificate or autopsy report alone (*n* = 1870), those with duplicate records (*n* = 135), those who did not have a diagnosis of NAFLD three or more years before HCC diagnosis (*n* = 37,147), and those with the other competing HCC etiologies listed above (*n* = 2432). After these exclusions, 1310 NAFLD-HCC cases were left for analysis. For controls, we identified 13,322 eligible population controls with unique records and without a prior history of cancer who reside in the same SEER catchment areas as the cases and were alive and 68 years or older, were not enrolled in HMO, not enrolled in Medicare because of end-stage renal disease or disability and had a diagnosis of NAFLD three or more years before the pseudo-diagnosis date. We then excluded those with the competing HCC etiologies (*n* = 3487), leaving 9835 cancer-free NAFLD controls for analysis. 

### 2.4. Assessment of Metabolic Risk Factors

By design, both the cases and controls had a diagnosis of NAFLD, and we examined the following metabolic risk factors in the setting of NAFLD: obesity, dyslipidemia, hypertension, hypothyroidism, type II diabetes mellitus, and metabolic syndrome. These conditions were identified from the Medicare Parts A and B claim files for the three years before HCC diagnosis (cases) or the three years before the pseudo-diagnosis date (controls). The ICD-9-CM and ICD-10-CM codes used to identify these conditions are listed in Appendix A [25,34,35,36,37]. We defined metabolic syndrome based on the US National Cholesterol Education Program Adult Treatment Panel III (NCEP ATP-III) recommendation as the presence of three or more of the following conditions: type II diabetes mellitus, obesity, hypertension, and dyslipidemia (elevated triglycerides or low high-density lipoprotein levels) [38]. We also determined a diagnosis of dyslipidemia, obesity, and cirrhosis based on ICD-9-CM and ICD-10-CM codes and determined participants’ smoking history using ICD codes (Appendix A) [39,40].

### 2.5. Statistical Analyses

Differences in participant demographics and pre-existing conditions were compared between the NAFLD-HCC cases and NAFLD controls using *t*-tests for continuous variables and Fisher’s exact tests for categorical variables. Multivariable-adjusted logistic regression was used to calculate odds ratios (ORs) and 95% confidence intervals, modeling HCC as the outcome and each metabolic risk factor separately as the predictor. Covariates were determined a priori, and we adjusted for the following covariates in all models: age (continuous), sex, race/ethnicity (White, Black, Hispanic, Asian, other), smoking history (yes/no), US geographic region (Midwest, Northeast, Southeast, Southwest, West), and dual enrollment in Medicare and Medicaid (low-income patients’ medical expense assistance program). We also calculated PAFs in multivariable models for the overall sample and by sex and cirrhosis status to estimate the proportions of HCC cases that could be theoretically eliminated should a particular risk factor be eliminated in the general population [41]. Because each predictor was modeled separately, we assessed multicollinearity among covariates using the variance inflation factor, which did not show significant collinearity between variables. All statistical tests were two-sided with a *p*-value < 0.05 considered statistically significant. Statistical analyses were performed using the R Statistical Software (version 4.1.2; R Foundation for Statistical Computing, Vienna, Austria).

## 3. Results

Distributions of demographic and baseline characteristics of the 1310 NAFLD-HCC cases and 9835 NAFLD controls are presented in Table 1. Briefly, the cases were on average one year older than controls, and the cases had a greater proportion of males (60%) compared to controls (31%). The cases were also more likely to have a smoking history (37%) than controls (26%), and there was a higher percentage of Hispanics and Asians among cases than controls. Among cases, the average time from NAFLD diagnosis to HCC was 6.7 years. 

### 3.1. Risk Factors, Odds Ratios, and PAFs for NAFLD-HCC

Table 2 shows differences in the distribution of the metabolic risk factors among the cases and controls. The NAFLD-HCC cases were more likely than controls to be obese, have a diagnosis of diabetes, and be classified as having metabolic syndrome. The cases were also more likely to have liver cirrhosis (51%) than controls (5%). However, no differences were observed in the baseline prevalence of dyslipidemia, hypertension, or hypothyroidism. 

In the multivariable-adjusted logistic regression models (Table 3), we found that type II diabetes mellitus (OR = 2.39, 95% CI: 2.04–2.79), metabolic syndrome (OR = 1.73, 95% CI: 1.49–2.01), and obesity (OR = 1.62, 95% CI: 1.43–1.85) were each associated with higher risk for HCC in persons with NAFLD, after adjusting for age, sex, race/ethnicity, smoking history, and dual Medicare and Medicaid enrollment. The highest PAR for HCC was observed among participants with pre-existing diabetes (42.1%, 95% CI: 35.7–48.5), followed by metabolic syndrome (28.8%, 95% CI: 21.7–35.9) and then obesity (13.2%, 95% CI: 9.6–16.8). No associations were found for dyslipidemia, hypertension, or hypothyroidism in the overall sample.

### 3.2. Stratified Analyses by Sex and Cirrhosis Status

In stratified analyses by sex, associations for metabolic risk factors seem to be slightly higher in women than men (Table 3). We found that among men, diabetes mellitus (OR = 2.19, 95% CI: 1.79–2.69), metabolic syndrome (OR = 1.57, 95% CI: 1.29–1.91), and obesity (OR = 1.48, 95% CI: 1.24–1.76) were associated with higher HCC risk. Among women, diabetes mellitus (OR = 2.71, 95% CI: 2.12–3.46), metabolic syndrome (OR = 1.95, 95% CI: 1.54–2.48), and obesity (OR = 1.80, 95% CI: 1.48–2.19) were again associated with higher HCC risk. PAR for HCC among men was 37.2% (95% CI: 28.7–45.7) for diabetes, 22.8% (95% CI: 13.6–32.1) for metabolic syndrome, and 9.8% (95% CI: 5.4–14.3) for obesity. Among women, PAR for HCC was 49.3% (95% CI: 39.7–58.9) for diabetes, 36.7% (95% CI: 25.5–47.9) for metabolic syndrome, and 18.0% (11.9–24.2) for obesity. Additionally, we found an inverse association between dyslipidemia and HCC risk among women (OR = 0.65, 95% CI: 0.47–0.89; PAR = −40.6, 95% CI: −72.4 to −8.7) but not men. As with the overall analyses, we did not find associations for hypertension or hypothyroidism in the stratified analyses by sex. 

We also performed stratified analyses by pre-existing cirrhosis status (yes, no), and the results were similar between the two groups (Table 4). Among persons with a prior history of liver cirrhosis, a higher risk for HCC was observed for those with diabetes (OR = 2.03, 95% CI: 1.48–2.79), metabolic syndrome (OR = 1.43, 95% CI: 1.05–1.95), and obesity (OR = 1.79, 95% CI: 1.36–2.34). The corresponding PAFs for HCC among cirrhotics with diabetes, metabolic syndrome, and obesity were 21.1% (95% CI: 11.8–30.4), 10.5% (95% CI: 1.6–19.3), and 8.5% (95% CI: 4.6–12.5), respectively. Diabetes, metabolic syndrome, and obesity were also associated with a higher risk for HCC among those without a history of cirrhosis, with ORs (95% CIs) of 2.04 (1.65–2.51), 1.68 (1.36–2.06), and 1.55 (1.30–1.86), respectively. The PAF for HCC among those with cirrhosis was 36.7% (95% CI: 27.3–46.1) for diabetes, 28.3% (95% CI: 18.1–38.6) for metabolic syndrome, and 12.0% (95% CI: 6.9–17.2) for obesity. There were no other associations observed. 

## 4. Discussion

In this large population-based study among Medicare participants in the US, we found that diabetes mellitus is the primary predisposing factor for HCC development in individuals with NAFLD, followed by metabolic syndrome and then obesity. We consistently observed this trend of association in subgroup analyses performed separately among men and women, and among participants with cirrhosis and those without cirrhosis. We also observed an inverse association between dyslipidemia and HCC risk among women only. However, no significant associations were observed for hypertension or hypothyroidism, which may be due to there being a similar distribution of these conditions among our NAFLD-HCC cases and the NAFLD controls. Collectively, the results suggest that HCC prevention strategies could be enhanced through focused prevention of diabetes, metabolic syndrome, and obesity. To our knowledge, this is the first study to investigate the metabolic risk factors of HCC in persons with NAFLD after careful exclusion of other competing risk factors of HCC.

Our study shows that individuals with diabetes have the highest population-attributable risk fraction for HCC development in our NAFLD population. Consistent with our findings, others have reported a higher HCC risk among NAFLD patients with diabetes compared with NAFLD patients without diabetes [23]. Other studies also suggest that glycemic control can reduce diabetes-associated HCC risk in patients with NAFLD [42]. Associations of metabolic syndrome and obesity with HCC risk have also been reported in previous studies [4,37]. Here, we provide evidence of the risk magnitudes and PAFs for these conditions in the setting of NAFLD, which is useful for population-based risk stratification in a clearly defined high-risk group. We also found an inverse association between dyslipidemia and HCC risk among women only. The reasons for this observation are not completely clear but could be due to greater adherence to the use of statins among women in our study cohort, sex differences in statin metabolism or efficacy, or some other factor, which warrants further investigation.

An unexpected finding of our study is the lack of association between hypertension and hypothyroidism with HCC risk. Previous studies have reported associations between these cardiometabolic conditions and HCC risk [35,37,43,44]. One explanation for the lack of association could be the similar prevalence of these conditions between our cases and controls, which may differ from other studies. Furthermore, the study participants were 68 years or older, and it is reasonable to expect that most of these participants would have these metabolic conditions since these conditions correlate positively with aging [45,46]. Therefore, verification of the findings in other populations with a much broader age range would be helpful. We also found that men had a higher prevalence of HCC (60%) than women, and diabetes, metabolic syndrome, and obesity were associated with higher HCC risk in both men and women. However, the magnitudes of risk and PAFs were slightly higher in women than men, pointing to a potentially stronger association between metabolic risk factors and HCC in women. 

NAFLD is a growing public health problem, with an estimated 25% of the world’s population having this liver condition [2,4]. In the US, the prevalence of NAFLD is estimated to be around 30% and is expected to continue to rise due to the growing obesity epidemic [2,8]. Although not all persons with NAFLD will develop HCC, the roughly 30% of NAFLD patients who are likely to progress to HCC within 10 years of NAFLD diagnosis will likely suffer severe morbidity owing to the often late stage of diagnosis, aggressive disease course, intractable symptoms, and physical limitations [5,25,36,47,48]. HCC is also a frequently fatal cancer, with about 80% of patients dying within 5 years of diagnosis [49]. Therefore, risk prevention remains the most effective strategy for reducing the morbidity and mortality associated with HCC.

The pathological pathway underlying the progression of NAFLD to HCC first involves the development of nonalcoholic steatohepatitis (NASH)—a more severe form of NAFLD marked by substantial hepatic inflammation and enlargement of hepatocytes [5,6,7,8]. NASH may then progress through various stages of liver fibrosis, from stage 0 (no liver stiffness) to stage 4 (significant liver stiffness) [50]. By fibrosis stage 4, most patients have already developed liver cirrhosis that can then progress to HCC [1,5,6,7]. However, many studies have shown that some NAFLD patients may develop HCC in the absence of cirrhosis [25,36,51,52,53]. While the exact proportion of patients with NAFLD who develop HCC without cirrhosis is not completely clear, studies have reported a wide variation in the proportion of NAFLD-related HCC cases that occur in the absence of cirrhosis, from as low as 3.4% to 42% in different populations [25,51,53,54,55]. The development of HCC without cirrhosis is more common in patients with NAFLD than in those with other HCC etiologies (e.g., viral hepatitis and alcohol abuse), and this may be explained by a complex interaction between nutrition, lifestyle, genetics, and epigenetic mechanisms [8,53,55,56]. In this study, 49% of our NAFLD-HCC cases did not have a prior history of liver cirrhosis. The roughly similar proportion of cirrhotics (51%) and non-cirrhotics in our study may explain the lack of difference in the results from stratified analyses by cirrhosis status. Importantly, our results suggest that diabetes, metabolic syndrome, and obesity could predispose to HCC development in persons with NAFLD in the presence or absence of cirrhosis. The potential molecular mechanisms underpinning the associations between diabetes, metabolic syndrome, and obesity with HCC risk are thought to involve adipokine signaling, the promotion of chronic hepatic inflammation, and hepatic metabolic dysregulation [1,57,58]. Importantly, lipotoxic liver injury and the associated release of pro-inflammatory cytokines appears to be a key molecular event that promotes the development of HCC in the setting of NAFLD [58,59]. Hence, chemo-preventive strategies targeting these mechanisms may help to ameliorate the deleterious progression of NAFLD.

The strengths of our study include the focus on metabolic risk factors of HCC in a clearly defined subgroup of at-risk patients. The use of the SEER-21 database, which covers roughly 37% of the US population [30], adds to the study’s strengths. The SEER-Medicare program links individual-level cancer diagnosis data with medical claims information for about 93% of all Medicare-eligible participants, making the data highly representative of individuals 65 years of age or older who live in the SEER catchment areas. The use of Medicare claims information for exposure assessment, as opposed to self-reporting, which is prone to recall bias, is also a strength. Furthermore, we used ICD-0-3 topographic and morphology codes and ICD-9/10-CM codes that have been vetted and recommended by an expert consensus panel [33] to identify the various metabolic conditions. We also had a large enough sample size to enable us to perform stratified analyses by sex and cirrhosis status, even after excluding other competing HCC risk factors such as viral hepatitis, alcoholic liver disease, autoimmune disorders, and rare genetic syndromes to optimally control for potential confounding. 

Our study was limited by the use of data on individuals 68 years of age or older, and the results may not be directly generalizable to younger populations. However, the median age of HCC diagnosis in the US is around 65 years [15,60]. Therefore, our results are largely generalizable to the highest-risk population. It is also known that an earlier age of HCC diagnosis is usually associated with a genetic predisposition, viral hepatitis infection, or alcoholic liver disease, conditions that are distinct from NAFLD-related HCC [2,14]. Metabolic conditions also tend to be more prevalent with aging [2,21], making a study population generally suitable for this investigation. We excluded individuals enrolled in HMOs because Medicare claims data do not include HMO claims. This may have introduced some level of selection bias but was necessary to avoid the misclassification of patients. While the ICD codes used in this study are commonly used in many studies, they are less granular than clinical data. We were also unable to assess the impact of hypoglycemic medication use on HCC risk among diabetics with NAFLD or differences in statin use between men and women in our study cohort. Lab values such as triglycerides were also not assessed in this study. For diabetes, there is some evidence that oral hypoglycemic medications may lower the HCC risk in NAFLD patients with co-existing diabetes [23]. However, additional, larger studies with detailed clinical and laboratory data as well as data on novel therapies for NAFLD/NASH are needed to address the limitations of our study. Given the observational nature of the study, we are unable to make causal inferences, and as with any study, there is the possibility of confounding by unmeasured factors. 

## 5. Conclusions

We examined the metabolic risk factors of HCC in individuals with a prior history of NAFLD after careful exclusion of other competing etiologies of HCC. We estimated, for the first time, the attributable population risk fractions for various metabolic conditions in the setting of NAFLD. The results show that type II diabetes mellitus has the highest attributable population risk fraction for HCC in patients with NAFLD, followed by metabolic syndrome and obesity. Given the increasing prevalence of NAFLD in the US and other developed and developing countries, controlling these metabolic conditions is likely to become critically important to the prevention of NAFLD-related HCC. Additional studies with detailed clinical and biochemical data on lipidomic and inflammatory markers would further improve our understanding of the metabolic risk factors of HCC in the setting of NAFLD.

## Figures and Tables

**Table 1 cancers-14-06234-t001:** Descriptive statistics of study participants (N = 11,145).

	NAFLD-HCC (N = 1310)	NAFLD-Controls (N = 9835)	*p*-Value
Age (years)	77.5 (5.8)	76.7 (6.6)	<0.001
Sex (Male)	786 (60.0%)	3063 (31.1%)	<0.001
Ethnicity			<0.001
White	935 (71.4%)	7735 (78.6%)	
Black	<35 †	562 (5.7%)	
Hispanic	214 (16.3%)	458 (4.7%)	
Asian	128 (9.8%)	670 (6.8%)	
Other	<35 †	410 (4.2%)	
Smoking history	484 (36.9%)	2534 (25.8%)	<0.001
Geographic location			<0.001
San Francisco—Oakland	30 (2.3%)	227 (2.3%)	
Connecticut	69 (5.3%)	451 (4.6%)	
Metropolitan Detroit	39 (3.0%)	327 (3.3%)	
Hawaii	42 (3.2%)	167 (1.7%)	
Iowa	55 (4.2%)	253 (2.6%)	
New Mexico	40 (3.1%)	171 (1.7%)	
Seattle (Puget Sound)	41 (3.1%)	250 (2.5%)	
Utah	35 (2.7%)	183 (1.9%)	
Metropolitan Atlanta	17 (1.3%)	135 (1.4%)	
San Jose—Monterey	26 (2.0%)	177 (1.8%)	
Los Angeles	95 (7.3%)	762 (7.7%)	
Greater California	164 (12.5%)	1325 (13.5%)	
Kentucky	90 (6.9%)	583 (5.9%)	
Louisiana	47 (3.6%)	370 (3.8%)	
New Jersey	99 (7.6%)	1057 (10.7%)	
Greater Georgia	89 (6.8%)	682 (6.9%)	
Idaho	18 (1.4%)	125 (1.3%)	
New York	212 (16.2%)	1941 (19.7%)	
Massachusetts	102 (7.8%)	649 (6.6%)	
Geographic US regions			<0.001
Midwest	94 (7.2%)	580 (5.9%)	
Northeast	482 (36.8%)	4098 (41.7%)	
Southeast	243 (18.5%)	1770 (18.0%)	
Southwest	40 (3.1%)	171 (1.7%)	
West	451 (34.4%)	3216 (32.7%)	
Medicare/Medicaid dual enrollment	357 (27.3%)	3558 (36.2%)	<0.001
Years from NAFLD diagnosis to HCC	6.7 (3.2)	-	

Continuous variables were summarized as mean (SD), while categorical variables were reported as frequency (percentage). *p*-values result from a two-sample *t*-test (continuous variables) or Fisher’s exact test (categorical variables). Abbreviations: HCC, hepatocellular carcinoma; NAFLD, nonalcoholic fatty liver disease. † Cell sizes with less than 35 counts for ethnicity and less than 11 counts for geographic region are suppressed according to the SEER-Medicare data use agreement.

**Table 2 cancers-14-06234-t002:** Comparison of metabolic risk factors and liver cirrhosis among NAFLD-HCC cases and NAFLD controls.

	NAFLD-HCC (N = 1310)	NAFLD Controls (N = 9835)	*p*-Value
**Metabolic conditions**			
Obesity	544 (41.5%)	3261 (33.2%)	<0.001
Dyslipidemia	1191 (90.9%)	9090 (92.4%)	0.16
Hypertension	1250 (95.4%)	9365 (95.2%)	0.84
Hypothyroidism	186 (14.2%)	1491 (15.2%)	0.39
Diabetes mellitus	1085 (82.8%)	6539 (66.5%)	<0.001
Metabolic syndrome	1051 (80.2%)	6887 (70.0%)	<0.001
**Other liver condition**			
Cirrhosis	668 (51.0%)	495 (5.0%)	<0.001

Continuous variables were summarized as mean (SD), while categorical variables were reported as frequency (percentage). *p*-values result from a two-sample *t*-test (continuous variables) or Fisher’s exact test (categorical variables). Abbreviations: HCC, hepatocellular carcinoma; NAFLD, nonalcoholic fatty liver disease.

**Table 3 cancers-14-06234-t003:** Associations between metabolic risk factors and NAFLD-HCC risk, and population-attributable risk fractions for each metabolic risk for the overall samples, stratified by sex.

	NAFLD-HCC Risk Compared to NAFLD Controls	
	OR (95% CI)	PAF (95% CI)
**Overall**		
Obesity	1.62 (1.43, 1.85)	13.2% (9.6, 16.8)
Dyslipidemia	0.73 (0.59, 1.11)	−19.1% (−23.4, 13.2)
Hypertension	1.03 (0.76, 1.38)	2.0% (−19.5, 23.5)
Hypothyroidism	1.06 (0.89, 1.26)	0.6% (−1.4, 2.6)
Diabetes mellitus	2.39 (2.04, 2.79)	42.1% (35.7, 48.5)
Metabolic syndrome	1.73 (1.49, 2.01)	28.8% (21.7, 35.9)
**Male**		
Obesity	1.48 (1.24, 1.76)	9.8% (5.4, 14.3)
Dyslipidemia	0.81 (0.60, 1.08)	−15.6% (−38.1, 6.9)
Hypertension	0.85 (0.59, 1.23)	−12.2% (−40.0, 15.5)
Hypothyroidism	1.05 (0.81, 1.36)	0.4% (−1.8, 2.5)
Diabetes mellitus	2.19 (1.79, 2.69)	37.2% (28.7, 45.7)
Metabolic syndrome	1.57 (1.29, 1.91)	22.8% (13.6, 32.1)
**Female**		
Obesity	1.80 (1.48, 2.19)	18.0% (11.9, 24.2)
Dyslipidemia	0.65 (0.47, 0.89)	−40.6% (−72.4, −8.7)
Hypertension	1.32 (0.80, 2.18)	21.2% (−12.7, 55.0)
Hypothyroidism	1.06 (0.84, 1.35)	0.9% (−2.9, 4.7)
Diabetes mellitus	2.71 (2.12, 3.46)	49.3% (39.7, 58.9)
Metabolic syndrome	1.95 (1.54, 2.48)	36.7% (25.5, 47.9)

Models were adjusted for age, sex, race, smoking, geographic region, and Medicare/Medicaid dual enrollment. No adjustment was made for sex in the sex-stratified models. Abbreviations: HCC, hepatocellular carcinoma; NAFLD, nonalcoholic fatty liver disease; PAF, population-attributable risk fraction.

**Table 4 cancers-14-06234-t004:** Multivariable-adjusted odds ratios (ORs) and 95% confidence intervals (CIs) for associations between metabolic risk factors and NAFLD-HCC risk, and population-attributable risk fractions for each metabolic risk by cirrhosis status.

	NAFLD-HCC Risk Compared to NAFLD-Controls	
	OR (95% CI)	PAF (95% CI)
**Cirrhosis**		
Obesity	1.79 (1.36, 2.34)	8.5% (4.6, 12.5)
Dyslipidemia	0.86 (0.43, 1.22)	−13.5% (−27.2, 14.2)
Hypertension	0.95 (0.52, 1.74)	−1.9% (−22.3, 18.5)
Hypothyroidism	1.01 (0.71, 1.42)	0.0% (−2.0, 2.1)
Diabetes mellitus	2.03 (1.48, 2.79)	21.1% (11.8, 30.4)
Metabolic syndrome	1.43 (1.05, 1.95)	10.5% (1.6, 19.3)
**No cirrhosis**		
Obesity	1.55 (1.30, 1.86)	12.0% (6.9, 17.2)
Dyslipidemia	1.02 (0.73, 1.43)	1.5% (−24.2, 27.3)
Hypertension	1.14 (0.74, 1.76)	10.2% (−21.4, 41.7)
Hypothyroidism	1.01 (0.79, 1.30)	0.1% (−2.6, 2.9)
Diabetes mellitus	2.04 (1.65, 2.51)	36.7% (27.3, 46.1)
Metabolic syndrome	1.68 (1.36, 2.06)	28.3% (18.1, 38.6)

Models were adjusted for age, sex, race, smoking, geographic region, and Medicare/Medicaid dual enrollment. No adjustment for sex in the sex-stratified models. Abbreviations: HCC, hepatocellular carcinoma; NAFLD, nonalcoholic fatty liver disease; PAF, population-attributable risk fraction.

## Data Availability

In compliance with the National Cancer Institute’s (NCIs) policy, the authors are not allowed to provide the SEER–Medicare data to other investigators. Investigators interested in using the SEER–Medicare data should apply to NCI SEER-Medicare Program (https://healthcaredelivery.cancer.gov/seermedicare/ (accessed on 16 February 2022).

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
