# Peer review of "Metabolic Risk Factors for Hepatocellular Carcinoma in Patients with Nonalcoholic Fatty Liver Disease: A Prospective Study"

_cancers, 2022, doi:10.3390/cancers14246234_

Round 1

Reviewer 1 Report

This is a ground-breaking large observational population study using the SEER-Medicare databse to examine metabolic risk factors in NAFLD leading to HCC.

It found that diabetes mellitus, followed by metabolic syndrome, then obesity, contributes significantly to the development of HCC in this population.

The study examines only patients 68 years or older with at least 3 years of NAFLD who eventually developed HCC in the period from 2003 to 2017. This database covers about 37% of the US population.

It identifies the risk factors for HCC in a quantitative manner: OR for DM 2.03-2.04, metabolic syndrome 1.43-1.68 and obesity 1.55-1.79, with or without cirrhosis.  It also finds no association between hypertension, dyslipidemia and hypothyroidism individually with HCC.

Reviewer 2 Report

In my opinion the paper can be informative and provide a valuable source document for anyone requiring a primer to know and understand this issue. But, in this manuscript there are numerous limitations and reasons why it cannot be considered for publication in this present form. One of the reasons for this is the lack of data about Ethical considerations. But, numerous shortcomings in the section Materials and Methods, Results and Discussion also make this paper not appropriate for publication in this form and significant corrections should be made (major revision). Some comments:        
  • Lines 80-90: Reconstruct this paragraph, so that the aim/aims of the study are emphasized, and that methodological steps are left for the next section `Materials and Methods`.  
  • Line 91: Add new subsections, firstly Study setting, Study design, Study population, Study Sample, Sample Size calculation, and then subsections Data Source, ..., Ethical consideration.  
  • Line 92: State whether the `Data Source` was publicly available, or whether a special permission was necessary to access and use data from this database.
  • Lines 94-95: In the stated claim which was backed up by two references (18 and 19) there is no word about the study design, but a description of the database was given. In this manuscript it is necessary to state the `Study design` that was used in this study.   
  • Lines 122-125: Add reference number for the IRB decision, that will confirm the statement in this sentence. 
  • Lines 127-150: State the inclusion and exclusion criteria for the cases more clearly. 
  • Lines 127-180: Explain the rationale for why 3 years were determined as the `cut point` in this study and back it up with an appropriate reference.
  • Lines 153-154: Can the statement in this sentence be correct - `The controls were subjected to the same inclusion and exclusion criteria as the cases.`. Explain. 
  • Lines 151-163: It is stated that this is a `random sample`. State what random sampling method was used. State whether you paid attention to the age and sex representation, which can be associated with both independent and with dependent variables, eg. outcome in this study. 
  • Lines 164-180: In order to increase readability, with data in this paragraph it is necessary to add one new Figure that will present the Flow chart in this study. 
  • Lines 181-195: In the way that you defined metabolic syndrome on Lines 189-192, define all risk factors/variables and correlates in this study, with citations of appropriate original references. It is not good practice to state in every paragraph of the section Materials and Methods `as has been done previously`, `used in many previous studies`, `have been described [18, 19].`. In Supplementary Table 1 the list with ICD codes as presented, but without definitions. Inscribe definitions for all risk factors/variables and correlates, with appropriate references in the subsection `Assessment of Metabolic Risk Factors`.    
  • Lines 196-211: State for which variables and by what criteria was adjusting done. 
  • Lines 196-211: State how was the collinearity between all variables in this study assessed (state by which test, state the results of the collinearity assessment, state how the issue of collinearity was handled in this manuscript). 
  • Lines 213-205: Is this paragraph necessary, in the section Results the results should be described, while the discussion should be left for the section Discussion. 
  • Lines 254-255: Explain why adjusting was not done, apart from the stated variables, for cirrhosis too. 
  • Lines 269-271: Check and describe what association was found for dyslipidemia in females. 
  • Lines 298-300: The claim in this sentence is not correct with regard to dyslipidemia. 
  • Lines 308-311: In the circumstances of such data, explain why 3 years were taken as the cut point in this study. 
  • Lines 315-337: Reconstruct the entire paragraph so that you discuss the associations with variables that were found in this study (obesity, diabetes, metabolic syndrome, as well as dyslipidemia in females), and not only cirrhosis that can be on the way of those variables towards the outcome in this study. 
  • Lines 338-339: Take into consideration the differences in females too, and consequently revise the stated claim. 
  • Lines 346-348: State where exactly in this manuscript have you said that `In this study, men had higher prevalence of HCC than women, ...`. Necessarily state what the prevalence of HCC was in men and prevalence of HCC in women. Discuss the differences you stated in this sentence. 
  • Lines 386-389: Inscribe the missing data.   

Reviewer 3 Report

Thank you for the opportunity to review this article. This prospective study aimed to assess the magnitudes of risk and population attributable risk fractions (PAF) associated with various metabolic conditions in relation to HCC risk. The authors examined associations between metabolic conditions and HCC development in individuals with a prior history of NAFLD. Precisely, the authors used the SEER-Medicare database, including 1.310 NAFLD-related HCC cases and 9.835 NAFLD controls in their analysis. The subjects with competing liver diseases (e.g., alcoholic liver disease and chronic viral hepatitis) were excluded, while baseline pre-existing diabetes mellitus, dyslipidemia, obesity, hypertension, hypothyroidism, and metabolic syndrome were assessed too. In addition, the authors investigated risk estimates by gender and liver cirrhosis status. The results suggest that diabetes, metabolic syndrome, and obesity were associated with higher HCC risk in individuals with NAFLD. The highest PAF for HCC was observed for pre-existing diabetes followed by metabolic syndrome and obesity.

Generally, good work has been done. Design and the methodology of the study, including the statistical approaches, are presented in details.

Overall, the authors should be commended for writing an interesting article. The tables are clear and useful to the readers. However, some tables (or figures) with biochemical data or inflammatory markers (if available) would be very nice and useful too. I recommend this article to be accepted after revision.

Here are some suggestions that authors might find useful:

1.     The authors used NCEP ATP-III to define metabolic syndrome. Why the updated document (Circulation 2009, 120, 1640-1645) was not used?

2.     Any comment regarding the relationship between lifestyle behavior and metabolic syndrome? Do lifestyle behavior may differ between different regions?

3.     In addition, were diet behaviors similar between all participants? Could it influence the findings?

4.     Criteria for dyslipidemia and obesity seem to be missing.

5.     Why biochemical data are completely missing? It would significantly enrich the manuscript.

6.     Similarly, whether some markers of inflammation are present in database? It is known that HCCs can be caused by lipotoxicity-mediated chronic inflammation, as the authors briefly mentioned. Availability of such data may further support your findings and contribute to better understanding of the underlying mechanisms. Furthermore, such markers may help in risk prediction and facilitate clinical translation of NAFLD-directed HCC chemoprevention. Please discuss this briefly when you are talking about molecular mechanisms underpinning the associations between diabetes, metabolic syndrome, and obesity with HCC risk.

7.     The authors stratified analyses by cirrhosis status. Was the presence of NASH analyzed too?

8.     The authors should emphasize what new is going to be added by their study given that there are a lot of recent publications on the same topic including some meta-analyses. Consequently, appropriate literature should be cited more comprehensively.

9.     The authors stated clearly both the strengths and limitations of the present study. However, concomitant therapies (such as statins, novel therapies for diabetes mellitus such as GLP-1RA, etc.), which could importantly influence cardiometabolic risk factors, were not mentioned? Is there any data about therapies that have been used? A brief discussion on this topic is highly recommended.

10.       Similarly, do the authors think that normal values of LDL-cholesterol, TG (used to diagnose metabolic syndrome) could be results of already used therapies, and, consequently some participants did not fulfill the criteria for the metabolic syndrome, but actually they have had this disease?

11.       In addition, potential use of novel therapies for diabetes could help not only to prevent the metabolic syndrome, but also to slow its progression when already present, as well as NAFLD and its progression. It would be useful to discuss this point too.

12.       The cases had a greater proportion of men (60%) compared to controls (31%). However, some results are different among gender. Please discuss this point.

13.       As authors stated that their results are largely generalizable to the highest risk population (older subjects), I suggest to highlight this point and eventually slightly change the title. It is very well known that the risk of cancer is associated with age, so a brief discussion about this could be useful.

14.       In the last paragraph of the subheading study population, there are some repetitions, please correct it.

15.       Conclusion is very general, all stated is already well known, so it should be specific highlighting the novelty of the present study. Also, further perspectives in this field may be mentioned.

Reviewer 4 Report

The present study aims to measure the risk associated with several general metabolic factors on HCC development from NAFLD using the Medicare databases. The study is well designed, presented and the conclusions are sound the problem is that the results obtained merely confirm previous obsservations, and relevant related studies are not cited, presented nor discussed on in the introduction nor in the discussion.

in sum, previous publications and data on the relevance of diabetes on HCC development from NAFLD should be included and discussed. Additionally, recent studies focus on hypoglucaemic treatments as a prevention tool for HCC development from NAFLD. Therefore, a sub analysis focused on the types of hypoglucaemic treatment and efficacy of the intervention on glucose homeostasis, would add novelty and interest to the study.

Round 2

Reviewer 2 Report

Thank you for the opportunity to re-review the manuscript ID: cancers-2019732. The authors have addressed all of the issues highlighted in my review, satisfactorily responded to my questions and provided explanations. The authors made the necessary changes to the manuscript. I believe that the changes they have made have significantly improved the manuscript. Thank you to the authors for their responses to my comments.   

Reviewer 3 Report

The authors have carefully addressed all my comments. I am satisfied with the improvements and I recommend the revised manuscript to be accepted.